# Individualized Contrast Media Application Based on Body Weight and Contrast Enhancement in Computed Tomography of Livers without Steatosis

**DOI:** 10.3390/diagnostics12071551

**Published:** 2022-06-25

**Authors:** Daan J. de Jong, Véronique V. van Cooten, Wouter B. Veldhuis, Pim A. de Jong, Madeleine Kok

**Affiliations:** Department of Radiology, University Medical Center Utrecht, Heidelberglaan 100, 3584 CX Utrecht, The Netherlands; d.j.dejong4@students.uu.nl (D.J.d.J.); v.v.vancooten@students.uu.nl (V.V.v.C.); w.veldhuis@umcutrecht.nl (W.B.V.); p.dejong-8@umcutrecht.nl (P.A.d.J.)

**Keywords:** computed tomography, liver attenuation, contrast media, personalized care

## Abstract

This study analyzes the homogeneity in liver attenuation of a body-weight-based protocol compared to a semi-fixed protocol. Patients undergoing abdominal multiphase computed tomography received 0.500 g of iodine (gI) per kilogram of body weight. Liver attenuation and enhancement were determined using regions of interest on scans in the pre-contrast and portal venous phases. The outcomes were analyzed for interpatient uniformity in weight groups. The subjective image quality was scored using a four-point Likert scale (excellent, good, moderate, and nondiagnostic). A total of 80 patients were included (56.3% male, 64 years, 78.0 kg) and were compared to 80 propensity-score-matched patients (62.5% male, 63 years, 81.7 kg). The liver attenuation values for different weight groups of the TBW-based protocol were not significantly different (*p* = 0.331): 109.1 ± 13.8 HU (≤70 kg), 104.6 ± 9.70 HU (70–90 kg), and 105.1 ± 11.6 HU (≥90 kg). For the semi-fixed protocol, there was a significant difference between the weight groups (*p* < 0.001): 121.1 ± 12.1 HU (≤70 kg), 108.9 ± 11.0 HU (70–90 kg), and 105.0 ± 9.8 HU (≥90 kg). For the TBW-based protocol, the enhancement was not significantly different between the weight groups (*p* = 0.064): 46.2 ± 15.1 HU (≤70 kg), 59.3 ± 6.8 HU (70–90 kg), and 52.1 ± 11.7 HU (≥90 kg). Additionally, for the semi-fixed protocol, the enhancement was not significantly different between the weight groups (*p* = 0.069): 59.4 ± 11.0 HU (≤70 kg), 53.0 ± 10.3 HU (70–90 kg), and 52.4 ± 7.5 HU (≥90 kg). The mean administered amount of iodine per kilogram was less for the TBW-based protocol compared to the semi-fixed protocol: 0.499 ± 0.012 and 0.528 ± 0.079, respectively (*p* = 0.002). Of the TBW-based protocol, 17.5% of the scans scored excellent enhancement quality, 76.3% good, and 6.3% moderate. Of the semi-fixed protocol, 70.0% scored excellent quality, 21.3% scored good, and 8.8% scored moderate. In conclusion, the TBW-based protocol increased the interpatient uniformity of liver attenuation but not the enhancement in the portal venous phase compared to the semi-fixed protocol, using an overall lower amount of contrast media and maintaining good subjective image quality.

## 1. Introduction

For the detection and characterization of hypovascular lesions, attenuation of the liver in the portal venous phase on contrast-enhanced computed tomography (CECT) is of importance. According to the literature, enhancement levels of 50 HU (40–70 HU) are considered to be diagnostic [1,2,3,4]. However, the eventual attenuation of the liver often varies between patients, which might result in different detectability of lesions. In the literature, different contrast media protocols have been described to achieve homogenous attenuation between patients [1,2]. Some advocate the use of fixed or semi-fixed protocols [5], whilst others have advocated the use of personalized protocols based on the total body weight (TBW) [1,6,7,8], lean body weight (LBW) [9,10,11,12,13,14,15], percentage of body fat [16], or body surface area [17,18]. Thus, there is no consensus on what protocol is best to use for what types of patients. One hypothesis is that solid organs have greater perfusion than adipose tissue and, therefore, a contrast protocol based on the lean body weight or fat-free mass can perform better than a protocol based on the total body weight in obtaining a more uniform enhancement between patients [11]. A recent retrospective study suggested that a semi-fixed protocol was not sufficiently effective in obtaining homogeneous attenuation between different weight groups, and the hypothesis existed that livers are enhanced most uniformly when an amount of contrast media based on the lean body weight is used [3]. However, a regression analysis showed that the added value of an LBW-based protocol compared to that of a solely TBW-based protocol was minor [3]. In addition, the administration of the amount of contrast media based on LBW is more time-consuming in daily clinical practice. It may, therefore, be sufficient to personalize the injection protocol based on body weight, and we adapted our CT protocols to this strategy in June 2020.

The aim of this study is to investigate the homogeneity of liver attenuation in patients who underwent a multiphase abdominal CT scan using a TBW-based contrast media injection protocol compared to a semi-fixed protocol. We hypothesize that a protocol wherein patients receive an individualized amount of contrast media per kilogram of TBW increases the homogeneity in attenuation between weight groups compared to a semi-fixed protocol.

## 2. Materials and Methods

### 2.1. Patients

All patients of 18 years or older who underwent a multiphase CT scan of the abdomen, including the portal venous phase, using an injection protocol based on TBW between 12 July 2020 and 8 June 2021 and who were not previously included in the semi-fixed protocol were included. A total of 116 patients were identified. Patients that met one of the following exclusion criteria were excluded: patients with liver cirrhosis (*n* = 5) [19], livers steatosis (liver parenchyma of <40 HU on unenhanced CT) (*n* = 8) [20], partial hepatectomy, extensive hepatic cystic lesions, technical problems during scanning, or artifacts on the CT due to stents or implants (*n* = 2); patients with characteristics not registered (*n* = 21); and patients who explicitly stated they did not wish to participate in medical research. The WMO, the Dutch Law on Medical Research, did not apply to this study as reviewed by the local medical ethical committee (METC, ref. 20-453/C). Informed consent was waived.

Newly obtained data from patients in the TBW-based protocol were compared to already-existing data on a semi-fixed protocol from a previous study (METC, ref. 20-025/C) [3].

### 2.2. Imaging Protocols

CT scans of the abdomen were performed on several CT scanners used in our clinic, i.e., an IQon Spectral CT, a Brilliance 64, an iCT 256 (Philips Healthcare, Best, The Netherlands), and a Somatom Force (Siemens Healthcare, Forchheim, Germany). For all the scans, the scan range for the unenhanced phase was the upper abdomen. The scan range for the portal venous phase was set from approximately 1 cm cranial of the diaphragm to the lower pelvis.

All the scans were performed with a tube voltage of 120 kV. The following parameters were used: for the IQon Spectral CT, 64 × 0.625 mm collimation, gantry rotation time of 0.27 s, quality reference tube currents of 37 mAs (unenhanced phase) and 116 mAs (portal venous phase), B (abdominal) kernel, and iDose level 3; for the Brilliance 64, 64 × 0.625 mm collimation, gantry rotation time of 0.4 s, quality reference tube currents of 73 mAs (unenhanced phase) and 102 mAs (portal venous phase), B (abdominal) kernel, and iDose level 3; for the iCT 256, 128 × 0.625 mm collimation, gantry rotation time of 0.4 s, quality reference tube currents of 72 mAs (unenhanced phase) and 127 mAs (portal venous phase), B (abdominal) kernel, and iDose level 3; and for the Somatom Force, 2 × 96 × 0.6 mm, gantry rotation time of 0.5 s, quality reference tube currents of 75 mAs (unenhanced phase) and 150 mAs (portal venous phase), Br40d kernel, and ADMIRE level 3.

A scan of the portal venous phase was acquired using bolus tracking. A circular region of interest (ROI) was placed in the abdominal aorta with a threshold of 150 HU. The post-threshold delay for scanning was 90 s for each scan.

### 2.3. Contrast Material and Injection Protocol

All the patients received an 18–20 G cannula in an antecubital vein and received (preheated to 37 °C) iodinated contrast media (Ultravist, Iopromide, 0.300 gI/mL; Bayer Healthcare, Berlin, Germany) [21]. The contrast media was injected using a standard, dual-head CT power injector (Stellant, Bayer Healthcare, Berlin, Germany).

According to the literature, the diagnostic enhancement is 50 HU (40–70 HU) [1,2,3,4]. Based on recent data, diagnostic enhancement can be achieved by the administration of 0.500 gI per kilogram of TBW. Therefore, our patients received 0.500 gI per kilogram of TBW with a fixed injection speed of 5.0 mL/s. To improve the efficiency of use in the clinic, the technicians rounded up the amount of contrast media slightly for some patients, with a maximum of 5%. The TBW and the height of the patients were asked verbally and registered in an electronic patient file.

The patients in the semi-fixed protocol received 36.0 gI for the group ≤70 kg, 45.0 gI for the 70–90 kg group, and 55.5 gI for the group ≥90 kg. The contrast media flow rates were 4.0 mL/s, 4.5 mL/s, and 5.0 mL/s, respectively [3]. Both protocols were followed by a saline flush of 50 mL following the contrast bolus at the same flow rate.

### 2.4. Quantitative and Subjective Image Analysis

The pre-contrast liver attenuation and post-contrast attenuation in the portal venous phases of patients in the TBW-based protocol were measured in Hounsfield Units (HU) by D.J.d.J. and V.V.v.C. (trained and supervised by M.K.) using circular ROIs of 1–2 cm in diameter. ROIs were placed in three different liver segments (Couinaud segments S2, S8, and S7), and the attenuation values were averaged. The enhancement was calculated by subtracting the liver attenuation on pre-contrast scans from the liver attenuation in the portal venous phase.

The subjective quality was scored by M.K. (with 6 years of experience in abdominal radiology). The subjective liver enhancement was scored on the liver attenuation scans. The liver attenuation was assessed using a four-point Likert scale (1 = excellent; 2 = good; 3 = moderate; and 4 = nondiagnostic).

For the quantitative and subjective analyses, the scorers were not blinded, as the data were compared to data obtained earlier [3].

### 2.5. Statistical Analysis

Statistical analyses were performed with R Statistical Software version 4.1.3. (R Foundation for Statistical Computing, Vienna, Austria). The normality of distribution was checked using histograms and Q–Q plots. Parametric data were given using the mean and standard deviation. Nonparametric data were given as medians and interquartile ranges. Since there was an uneven distribution of sex in the protocols, nearest neighbor propensity score matching without replacements was performed based on patient gender, age, and weight. Thereafter, the patient characteristics of the protocols were compared using a Fisher’s exact test to compare the distribution of patient sex, as well as either an independent *t*-test for parametric data or Mann–Whitney U tests for nonparametric data. An interscorer agreement was determined for the enhancement values as scored by the scorers. For this, a single-measure, two-way, mixed, intraclass correlation coefficient (ICC) for consistency was used. Given the excellent agreement (ICC > 0.90), the scores of both readers were averaged. The enhancement and liver attenuation were compared using independent *t*-tests. A one-way ANOVA with Tukey post hoc analysis was used to analyze differences in the enhancement and post-contrast liver attenuation between the weight classes. The variance in enhancement and liver attenuation was compared using Levene’s test, a test that compares the mean absolute deviations to the means between groups. *p*-values of *p* ≤ 0.05 indicated a significant difference.

## 3. Results

In the propensity-score-matching, 160 patients were matched out of the 80 patients in the TBW-based protocol (55.2% male, 78.0 kg (IQR: 68.0–89.0)) and the 102 patients in the semi-fixed protocol (70.6% male, 81.0 kg (IQR: 72.8–90.0)). The standard mean difference of the propensity-score-matching for all the patients was 0.54; for matched patients, this was reduced to 0.30.

### 3.1. Baseline Characteristics

There were 80 patients in the TBW-based cohort (56.3% male) and 80 patients in the semi-fixed cohort (62.5% male) (*p* = 0.260). The patients in the TBW-based cohort had a median age of 64 years. In the semi-fixed cohort, there were slightly younger patients, with a median age of 63 years (*p* = 0.040). The median TBWs of the TBW-based and semi-fixed protocols were not significantly different: 78.0 kg versus 81.7 kg (*p* = 0.430), respectively. The mean body mass index (BMI) was 26.8 kg/m^2^ for the TBW-based protocol and 26.5 kg/m^2^ for the semi-fixed protocol (*p* = 0.715). For the TBW-based and semi-fixed protocols, average administered iodine amounts of 40.2 gI and 42.1 gI (*p* = 0.074) per scan were used, and the amounts of iodine per kg of TBW were 0.499 gI for the TBW-based protocol and 0.528 gI for the semi-fixed protocol (*p* = 0.002). The full details are presented in Table 1.

### 3.2. Objective Liver Attenuation and Enhancement

The ICC value for enhancement between the two scorers was excellent (ICC = 0.982 (*p* ≤ 0.001)).

The average pre-contrast liver attenuation values for the TBW-based and semi-fixed groups were 58.2 HU and 56.4 HU (*p* = 0.085), respectively. For the separate weight categories, the attenuation values for the TBW-based and semi-fixed groups, respectively, were 60.8 HU and 61.6 HU for the ≤70 kg groups (*p* = 0.572), 59.3 HU and 55.9 HU for the 70–90 kg groups (*p* = 0.009), and 52.5 HU and 52.5 HU for the ≥90 kg groups (*p* = 0.999). Further details can be found in Table 2.

The average post-contrast liver attenuation values for the TBW-based and semi-fixed protocols were 105.9 HU and 110.6 HU (*p* = 0.012), respectively. For the separate weight categories, the attenuation values for the TBW-based and semi-fixed protocols, respectively, were 109.1 HU and 121.1 HU for the ≤70 kg groups (*p* = 0.009), 104.6 HU and 108.9 HU for the 70–90 kg groups (*p* = 0.053), and 105.1 HU and 105.0 HU for the ≥90 kg groups (*p* = 0.970), respectively (see also Table 2 and Figure 1).

The mean liver enhancements for the TBW-based and semi-fixed protocols were 46.7 HU and 54.2 HU (*p* < 0.001), respectively. For the separate weight categories, the enhancement values for the TBW-based and semi-fixed protocols, respectively, were 46.2 HU and 59.4 HU for the ≤70 kg groups (*p* = 0.006), 59.3 HU and 53.0 HU for the 70–90 kg groups (*p* < 0.001), and 52.1 HU and 52.4 HU for the ≥90 kg groups (*p* = 0.915). Full details are presented in Table 2 and Figure 2.

### 3.3. Comparison of Post-Contrast Liver Attenuation and Enhancement

The mean liver attenuation values between the weight groups of the TBW-based protocol were not significantly different (*p* = 0.331). The mean liver attenuation values between the weight groups of the semi-fixed protocol were significantly different (*p* < 0.001). For this protocol, the post hoc analysis showed that the ≤70 kg group’s liver attenuation was higher than the attenuations of the 70–90 kg (*p* = 0.001) and ≥90 kg (*p* < 0.001) groups, as shown in Table 2. The variance between the two protocols in liver attenuation was not significantly different (*p* = 0.465).

The mean liver enhancement values of the subgroups of the TBW-based (*p* = 0.064) and semi-fixed (*p* = 0.069) protocols were not significantly different. The variance between the two protocols in enhancement was not significantly different (*p* = 0.593).

### 3.4. Subjective Quality

Of the 80 scans obtained in the TBW-based protocol, 17.5% (14/80) of the scans were scored as excellent, 76.3% (61/80) were scored as good, 6.3% (5/80) were scored as moderate, and no scans were scored as nondiagnostic. Of the ≤70 kg group, 23.8% had excellent enhancement (5/21), 66.7% (13/21) had good enhancement, and 9.5% (3/21) had moderate enhancement. Of the 70–90 kg group, 14.6% (6/41) of the scans were scored as excellent, 78.0% (32/41) were scored as good, and 7.3% (3/41) were scored as moderate. Of the ≥90 kg group, 16.7% (3/18) were scored as excellent, and 83.3% (15/18) were scored as good.

Of the 80 matched patients in the semi-fixed protocol, 70.0% (56/80) were scored as excellent, 21.3% (17/80) were scored as good, 8.8% (7/80) were scored as moderate, and no scans were scored as nondiagnostic. Of the ≤70 kg group, 87.5% had excellent enhancement (14/16), and 12.5% (2/16) had moderate enhancement. Of the 70–90 kg group, 73.5% (36/49) of the scans were scored as excellent, 20.4% (10/49) were scored as good, and 6.1% (3/49) were scored as moderate. Of the ≥90 kg group, 40.0% (6/15) were scored as excellent, 46.7% (7/15) were scored as good, and 13.3% (2/15) were scored as moderate.

## 4. Discussion

Liver attenuation was more uniform when contrast media was given in a per-patient protocol based on TBW compared to a semi-fixed protocol. Most patients showed good to excellent enhancement, subjectively, under both strategies. Patients that were enhanced relatively moderately were in the lowest weight category of the TBW-based protocol, indicating that these patients needed to be dosed slightly higher. Overall, the TBW-based protocol used less contrast media whilst maintaining overall good image quality with more uniform attenuation after contrast.

Our results are comparable to the results of Perrin et al. [7] and Martens et al. [22], who also compared fixed or semi-fixed protocols to weight-based protocols. The study of Perrin et al. [7] was a noninferiority trial wherein patients were scanned twice, both in a semi-fixed protocol based on groups of TBW and in a protocol based on a per-patient amount of 0.490 gI per kilogram of TBW. Based on the liver attenuation values, the TBW-based protocol performed noninferior to the semi-fixed protocol whilst also using less contrast media and maintaining image quality. They suggested that a TBW-based protocol should be favored. In the study by Martens et al. [22], a protocol with a fixed amount of iodine was compared to a protocol with 0.400 gI per kilogram of TBW at 90 kV. The patients had comparable weights to the patients in our study, and the study by Martens et al. analyzed the liver attenuation of the liver in comparable weight groups to our study. They also found more homogeneous attenuation between different weight groups whilst using less contrast media when using a TBW-based protocol. Overall, based on previous research and our findings, TBW-based protocols were noninferior or better in achieving homogeneous attenuation of the liver.

In this study, the lightweight, ≤70 kg group of patients in the TBW-based protocol received 0.118 gI per kilogram TBW, which was less compared to the lightweight patients in the semi-fixed group, and the enhancement was somewhat lower in this group. Lightweight patients in the semi-fixed protocol received the highest amount of iodine per kilogram of TBW (0.619 gI per kilogram TBW) and, subsequently, the attenuation levels were highest for this group. Reassuringly, for the TBW-based protocol, no nondiagnostic scans were found. This suggests that patients who received an amount of 0.619 gI per kilogram TBW in the semi-fixed protocol were somewhat overdosed. Furthermore, the difference between the lightweight patients in the semi-fixed group and the heaviest weight group in the semi-fixed group was 0.172 gI per kilogram TBW, and the attenuation values showed significant differences between those weight groups, while the subjective image quality was maintained as diagnostic for both weight groups. These data showed that the semi-fixed protocol probably overestimated the amount of iodine for lighter-weight patients, and the amount of iodine could generally be reduced.

Despite the positive finding that the amount of contrast media was likely overestimated for lightweight patients in the semi-fixed protocol, care is needed not to underdose lightweight patients with a TBW-based protocol. Our results showed that the amount of 0.500 gI per kilogram of TBW, which is in line with literature for 120 kV scans [1,8,23], might not be enough for lightweight patients. Different trends were observed for the TBW-based protocol compared to the semi-fixed protocol: contrary to the enhancement values in the semi-fixed protocol, in the TBW-based protocol, the mean enhancement increased as weight increased. For patients in the TBW-based ≤70 kg group, an average of 0.501 gI per kilogram resulted in an enhancement of 46.2 HU compared to 59.4 HU using 0.619 gI per kilogram in the semi-fixed protocol. The 70–90 kg group in the TBW-based protocol was enhanced to 44.5 HU, less than the 53.0 HU of the semi-fixed group, using 0.499 gI and 0.523 gI per kilogram of TBW, respectively. The highest weight group reached diagnostic enhancement and were not enhanced significantly different using 0.498 gI. The results in this study showed that 0.500 gI per kilogram of TBW was not enough to reach diagnostic enhancement for patients in the ≤70 kg and 70–90 kg groups, according to the objective cut-off of 50 HU described in the literature [1,2,3,4]. Moreover, the subjective image analysis of the scans in the portal venous phase in our study showed that most scans showed excellent or good enhancement and that, in our population, overall 0.500 gI per kilogram of TBW was enough to reach excellent and good quality scans. Nevertheless, the overall subjective enhancement of scans in the TBW-based protocol more often were scored as good quality than as excellent, as compared to the semi-fixed protocol. In particular, the quality of enhancement of the ≤70 kg and 70–90 kg groups was subjectively scored less compared to the semi-fixed protocol, as well as the ≥90 kg group in the same protocol. For the ≥90 kg groups, scans in the TBW-based protocol scored better; no scans were scored as moderate in the TBW-based protocol, whereas this was the case for 13.3% in the semi-fixed protocol. Overall, the results from the subjective analysis and objective enhancement suggest that patients in the lightweight ≤70 kg and 70–90 kg groups received too little contrast media to reach diagnostic enhancement, and a higher amount of contrast media per kilogram of TBW needs to be considered for these groups.

Furthermore, although analysis of the enhancement of the separate groups in the semi-fixed protocol showed more uniform attenuation, the overall variances of enhancement and liver attenuation were not significantly different between the two protocols. This may have been caused by the semi-fixed administration of contrast media in three groups based on weight. The differences may, therefore, be smaller than studies such as the study of Martens et al. [22] comparing a fixed protocol, rather than a semi-fixed protocol, to an individualized protocol. Nonetheless, an analysis comparing the attenuation of the different weight groups showed that the interpatient uniformity increased using a TBW-based protocol compared to a semi-fixed protocol.

Some study limitations warrant future consideration. Firstly, in this study, it may have been possible that undetected steatosis altered the results. Steatosis was defined in this study as a nonenhanced liver of <40 HU, and subsequently, patients were excluded from the analysis because steatotic livers enhance differently than nonsteatotic livers [20]. However, steatosis is known to be a pre-developing disease [24,25] and the sensitivity of the <40 HU cutoff is low [26]. Moreover, the attenuation values of livers with steatosis (<40 HU) were unknown due to our exclusion criteria. Secondly, the setup of this study was that of a nonrandomized, single-center study, where we compared newly obtained data to already-existing data. Therefore, the scorers of the objective and subjective analyses could not be blinded to the injection protocol. Moreover, two separate groups of patients were included in two separate periods, introducing the possibility of confounding and bias. We countered this problem using propensity-score-matching. Lastly, in this study, relatively small groups of patients were analyzed. The lowest and highest weight groups were smaller than the middle weight group. A future randomized, controlled trial using larger amounts of patients and equal amounts of patients in all the weight groups can best be used to confirm the results from this trial.

## 5. Conclusions

In conclusion, a TBW-based protocol increased the interpatient uniformity of liver attenuation but not the enhancement in the portal venous phase between different weight groups compared to a semi-fixed protocol, using an overall lower amount of contrast media while maintaining subjective diagnostic image quality. The lightweight patients showed lower mean enhancement values, which might imply that lightweight patients should receive a slightly higher dose of contrast media per kilogram of TBW.

## Figures and Tables

**Figure 1 diagnostics-12-01551-f001:**
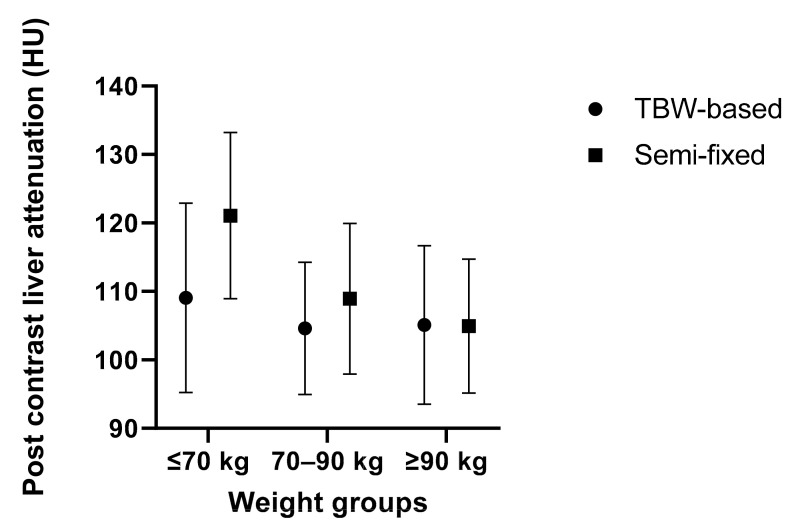
Post-contrast liver attenuation: means and standard deviations of post-contrast liver attenuation (HU) for TBW-based and semi-fixed protocols divided into subgroups of ≤70 kg, 70–90 kg, and ≥90 kg.

**Figure 2 diagnostics-12-01551-f002:**
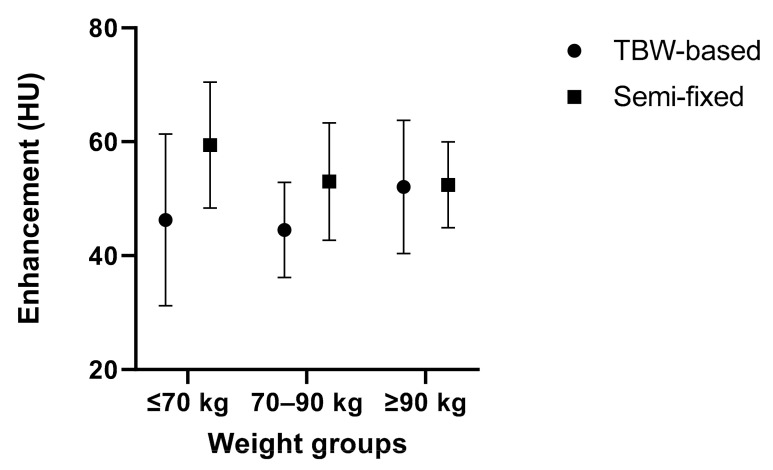
Liver enhancement: means and standard deviations of liver enhancement (HU) for TBW-based and semi-fixed protocols divided into subgroups of ≤70 kg, 70–90 kg, and ≥90 kg.

**Table 1 diagnostics-12-01551-t001:** Baseline characteristics.

		TBW-Based	Semi-Fixed	
**Total**	No. participants	80	80	
	Male	56.3%	62.5%	0.260
	Age (years)	64 (50–70)	63 (55–73)	**0.040**
	TBW (kg)	78.0 (68.0–89.5)	81.7 (72.5–89.0)	0.430
	BMI	26.8 (±4.7)	26.5 (±4.3)	0.715
	Grams of iodine	40.2 (±7.9)	42.1 (±4.4)	0.074
	Grams of iodine/kgTBW	0.499 (±0.012)	0.528 (±0.079)	**0.002**
**Group ≤ 70 kg**	No. participants	21	16	
	Male	10%	31%	0.107
	Age (years)	61 (52–69)	66 (56–75)	0.291
	TBW (kg)	63.0 (59.0–67.0)	62.5 (55.9–64.5)	0.308
	BMI	22.9 (±2.7)	21.5 (±2.0)	0.108
	Grams of iodine	31.4 (±2.00)	37.7 (±3.1)	**0.000**
	Grams of iodine/kgTBW	0.501 (±0.008)	0.619 (±0.061)	**0.000**
**Group 70–90 kg**	No. participants	41	49	
	Male	66%	69%	0.447
	Age (years)	64 (50–69)	64 (54–72)	0.201
	TBW (kg)	79.0 (74.0–83.0)	82.0 (78.9–85.1)	0.095
	BMI	26.6 (±3.0)	26.4 (±2.6)	0.808
	Grams of iodine	39.6 (±3.1)	42.4 (±3.8)	**0.000**
	Grams of iodine/kgTBW	0.499 (±0.012)	0.523 (±0.054)	**0.006**
**Group ≥ 90 kg**	No. participants	18	15	
	Male	89%	73%	0.242
	Age (years)	56 (49–70)	62 (56–67)	0.281
	TBW (kg)	104 (95.0–115)	101 (94.2–110)	0.789
	BMI	31.8 (±5.2)	31.8 (±4.2)	0.985
	Grams of iodine	52.0 (±4.3)	45.8 (±3.8)	**0.000**
	Grams of iodine/kgTBW	0.498 (±0.014)	0.447 (±0.062)	**0.002**
**Mean (±SD) or Median (IQR)**				***p*-value**

Baseline characteristics of the included patients for the TBW-based and semi-fixed protocols. Normally distributed data are given as means with standard deviation and nonparametric data are given as medians with interquartile ranges. Fisher’s exact test was used to compare the distribution of patient sex, and either an independent *t*-test for parametric data or Mann–Whitney U tests for nonparametric data were used to compare other patient characteristics. Bold indicates a statistically significant difference. TBW = total body weight; BMI = body mass index.

**Table 2 diagnostics-12-01551-t002:** Enhancement and liver attenuation.

	TBW-Based			Semi-Fixed		
	Pre-contrast attenuation	Enhancement	Post-contrast attenuation	Pre-contrast attenuation	Enhancement	Post-contrast attenuation
**Total**	58.2 (±7.1)	46.7 (±11.5)	105.9 (±11.3)	56.4 (±5.6)	54.2 (±10.2)	110.6 (±12.2)
**Group ≤ 70 kg**	60.8 (±5.2)	46.2 (±15.1)	109.1 (±13.8)	61.6 (±3.9)	59.4 (±11.0)	121.1 (±12.1)
**Group 70–90 kg**	59.3 (±6.8)	44.5 (±8.4)	104.6 (±9.67)	55.9 (±5.1)	53.0 (±10.3)	108.9 (±11.0)
**Group ≥ 90 kg**	52.5 (±7.2)	52.1 (±11.7)	105.1 (±11.6)	52.5 (±4.7)	52.4 (±7.5)	105.0 (±9.79)
Intergroup comparison		0.069	0.331		0.064	**0.000**
**Mean (±SD)**						***p*-value**

Mean and standard deviation of enhancement and pre- and post-contrast liver attenuation (HU) for the TBW-based and semi-fixed protocols in total and divided into subgroups of ≤70 kg, 70–90 kg, and ≥90 kg. The intergroup comparison was conducted using a one-way ANOVA. Results from the post hoc analysis are described in the text. Bold indicates a statistically significant difference.

## Data Availability

The data presented in this study are available on request from the corresponding author. The data are not publicly available due to ongoing unpublished research.

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
