# Peer review of "Individualized Contrast Media Application Based on Body Weight and Contrast Enhancement in Computed Tomography of Livers without Steatosis"

_diagnostics, 2022, doi:10.3390/diagnostics12071551_

Round 1

Reviewer 1 Report

It is a nice presented manuscript. I have some questions about your manuscript.

1.     According to your opinion, TBW-based protocol is one of the best methods for evaluating the liver attenuation and so on. Even though the body weight is the same, the distribution of body composition can be different. In order to adjust for these different conditions of body composition, a body shape index (ABSI) is recommended. What is your opinion about ABSI and ABSI_Z ?

2.     I think that the verbal questionnaire is incorrect method for obtaining the results of the TBW and height of the patients.

Thank you.

Reviewer 2 Report

This study analyzed the homogeneity in liver attenuation of a body weight-based protocol
(TBW.based) compared to a semi-fixed protocol. The mean administered amount of iodine per kilogram was less for the TBW-based compared to the semi-fixed protocol, 0.499±0.012 and 0.528±0.079, respectively (p=0.002). Of the TBW-based protocol, 17.5% of scans scored excellent enhancement quality, 76.3% good, and 6.3% 27
moderate. Of the semi-fixed protocol, 70.0% scored excellent quality, 21.3% scored good, and 8.8% scored moderate.

The paper is well written and easy to read.

Minor points: The Authors should discuss whether TBW based protocol scored worse than semi-fixed protocol.
